# Surface Properties of Retrieved Cementless Femoral Hip Endoprostheses Produced from a Ti6Al7Nb Alloy

**Klemen Avsec [1], Monika Jenko [2,3,\*], Marjetka Conradi [3], Aleksandra Kocijan [3], Alenka Vesel [4], Janez Kovač [4], Matjaž Godec [3], Igor Belič [3], Barbara Šetina Batič [3], Črtomir Donik [3], Matevž Gorenšek [5], Boštjan Kocjančič [1] and Drago Dolinar [1,2]**

1   Department of Orthopedic Surgery, University Medical Centre, Zaloška 9, 1000 Ljubljana, Slovenia;
    kavsec@gmail.com (K.A.); kocjancicb@gmail.com (B.K.); dolinardrago@gmail.com (D.D.)
2   MD-RI Institute for Materials Research in Medicine, Bohoričeva 5, 1000 Ljubljana, Slovenia
3   Institute of Metals and Technology, Lepi pot 11, 1000 Ljubljana, Slovenia; marjetka.conradi@imt.si (M.C.);
    aleksandra.kocijan@imt.si (A.K.); matjaz.godec@imt.si (M.G.); igor.belic@imt.si (I.B.);
    barbara.setina@imt.si (B.Š.B.); crtomir.donik@imt.si (Č.D.)
4   Jožef Stefan Institute, Jamova 39, 1000 Ljubljana, Slovenia; alenka.vesel@ijs.si (A.V.); janez.kovac@ijs.si (J.K.)
5   MD Medicina, Bohoričeva 5, 1000 Ljubljana, Slovenia; matevz.gorensek@md-medicina.si
\*   Correspondence: monika.jenko@imt.si

**Abstract:** We have investigated new and retrieved cementless hip endoprostheses that prematurely failed due to (i) aseptic loosening, (ii) infection and (iii) latent infection. The aim was to better understand the physico-chemical phenomena on the surfaces and sub-surfaces of the Ti6Al7Nb alloy implant. The results of our studies should enable us to distinguish the causes of premature failure, optimize the surface modification, achieve optimal osseointegration and extend the useful lifetime of the implants. The surface properties of the Ti6Al7Nb alloys of the hip-stem endoprostheses (30 retrieved and 2 new) were determined by contact-angle measurements and the average surface roughness. The surface chemistry and microstructure were analysed by scanning electron microscopy (SEM) for morphology, energy-dispersive X-ray spectroscopy (EDS) for the chemistry, and electron back-scatter diffraction (EBSD) for the phase analysis; Auger electron spectroscopy (AES) and X-ray photoelectron spectroscopy (XPS) for the surface chemistry; and electrochemical measurements for the corrosion. The improved wettability of the grit-blasted surface of the Ti6Al7Nb stems after autoclaving was measured, as was the super wettability after oxygen-plasma sterilization. The secondary-electron images showed that the morphology and microstructure of the new and retrieved stems (prematurely failed due to aseptic loosening, infection and latent infection) differ slightly, while the EDS analysis revealed corundum contamination of the grit-blasted surface. We found corundum-contaminated Ti6Al7Nb stem surfaces and sub-surfaces for all the investigated new and retrieved implants. These residues are a potential problem, i.e., third-body wear particles, and probably induce the osteolysis and aseptic loosening.

**Keywords:** Ti6Al7Nb alloy; premature hip endoprostheses failure; osseointegration; corundum contamination; corrosion; scanning electron microscopy (SEM); energy-dispersive X-ray spectroscopy (EDS); Auger electron spectroscopy (AES); X-ray photoelectron spectroscopy (XPS)

## 1. Introduction

Total hip-joint arthroplasty (THA) is the most successful surgical method for relieving pain, correcting deformities and treating degenerative joint disease and trauma. The number of patients who need a total joint arthroplasty, and possible revision, is increasing due to an aging population.

Kurtz et al. reported that the predictions for the primary revision of hip and knee arthroplasty in the USA in the next 10 years is of 1 million total joint arthroplasties (TJAs) per year and 2.5 million THA per year worldwide [1]. The (TJA) replacement surgeries have been very successful for decades, less than 10% of these implants fail prematurely in the first 10–20 years, causing several additional patients annually [2]. Aseptic loosening and periprosthetic joint infection are the main causes for the premature failure of joint arthroplasty [3–7]. The process of an ageing population continues, life expectancy is increasing, so besides premature failures, many patients are outliving their implants, which is a second concern. Both factors lead to the prediction of a large increase in implant failure in the near future and an enormous increase in health costs. Therefore, it is necessary to reduce the number of revisions to reduce the costs. On the market there are approximately 60 different types of single-stem cementless prostheses and knowledge of the behaviour of the individual prosthesis in certain clinical conditions is very important [2].

Aseptic loosening can be induced by different causes, such as the micromotion of the implant in the bone during loading, corundum wear particles from grit-blasted surfaces, wear-particle generation, causing inflammation and bone resorption, and consequently the formation of a poor functional interface between the implant and the patient's bone [8]. Infections of the implant surface occur when bacteria attach to the implant surfaces, and form a biofilm, hindering their elimination [8,9]. Costerton et al. report that particulate bacteria species are responsible for the majority of pathogens. *Staphylococcus aureus* and *epidermidis* cause almost 70% of orthopaedic implant infections [9,10].

The aim of our studies was to better understand the physico-chemical phenomena on the surfaces and sub-surfaces of the Ti6Al7Nb alloy cementless hip endoprostheses. The results of our studies should enable us to distinguish the causes of premature failure, optimize the surface modification, achieve optimal osseointegration, and extend the useful lifetime of the implants.

Knowledge of the behaviour of individual prosthesis in certain clinical conditions is very important. We have selected for detailed investigation the cementless Zweymuller-type hip endoprostheses, new and retrieved, that prematurely failed due to aseptic loosening, infection and latent infection.

The Ti6Al4V alloy, as we already described in our previous paper [5], has been the most popular metallic implant material used in THA (and is still widely used in the USA), but in recent decades Ti6Al7Nb has become a very popular material because of its better biocompatibility due to the replacement of vanadium (toxic when released over certain critic levels) with niobium. The Ti6Al7Nb alloy is now the most commonly used implant material for the femoral stems of hip and knee endoprostheses [4–7].

The surface roughness of the cementless stem of a hip endoprosthesis is very important for good osseointegration. With the aim of improving the osseointegration of orthopaedic implants, many surface-modification strategies have been developed, focusing on the biomaterial surface properties and health costs [6–13]. Corundum grit blasting is the most widely used mechanical surface treatment for roughening the surfaces of titanium cementless implants due to its good surface-roughening properties. Corundum-grit blasting introduces contamination of the surface of the Ti6Al7Nb alloy implant by retained corundum, which can adversely affect the osseointegration process of the implant. This is in addition to the roughening effects and the source of the wear particles that could cause debris-related aseptic loosening [14–18].

The retrieved biomaterial devices, i.e., the metallic components of hip endoprostheses, that are stored for the investigation of premature failures, etc., must be cleaned and sterilized. The cleaning procedure removes or reduces the visible contamination (blood, protein and debris) on the surface, while the sterilization stops the reproduction of micro-organisms, bacteria, spores and fungi [19–24].

The cleaning and sterilization influence the properties of new and retrieved Ti6Al7Nb alloy surfaces, which were studied to find possible changes in terms of wettability, roughness, morphology, microstructure, corrosion and surface chemistry [25–29].

The main aim of the investigation was to obtain data about the behaviour of this material and find out more about the very complex phenomena of premature failure due to aseptic loosening that results from the wear-debris particles originating from the grit-blasted surface.

## 2. Materials and Methods

### 2.1. Implants/Stems

The retrieved Ti6Al7Nb stems of the cementless ZweiMüller (ZM) type of hip endoprostheses were selected from revision surgeries performed at the Orthopedic Clinic of the University Medical Centre (UMC) Ljubljana (Ljubljana, Slovenia). We also investigated two new stems (after their expiry date) for comparison.

For a preliminary investigation, 30 stems of cementless hip endoprostheses that prematurely failed due to (i) aseptic loosening (10 implants), (ii) infection (10 implants) and (iii) latent infection (10 implants) were selected. The interval between the primary hip replacement and the revision surgery was 36 to 239 months for aseptic loosening, 3 to 36 months for infection and 36 to 168 months for latent infection. The retrieved stems were ZM type from an unknown manufacturer, while the new ones were Smith and Nephew (London, UK) Ti6Al7Nb stems of cementless hip endoprostheses.

After the revision surgery, the retrieved implants were sent for sonication and microbiological analyses in Ringer's solution and afterwards for cleaning and sterilization. All the retrieved stems were cleaned according to standard procedures at UMC Ljubljana, which consist of immersion in 2% microsoap solution, followed by acetone, isopropanol ($xN$), 95% ethanol ($xN$), and deionized water ($xN$); ($xN$) is the number of repeated processes. Sterilization was performed by autoclaving according to a standard protocol at 120 °C and a pressure of 1.25 bar for 20 min. One of the stems was sterilized by gaseous oxygen plasma for comparison to find out whether this method is compatible with a standard autoclaving sterilization procedure. Afterwards, sterilized stems were kept in sterile bags in a dry place for further investigations. New femoral components were cleaned and sterilized at the manufacturer's site and the special bags were opened on site just before the implantation. The cleaning and sterilization processes, according to the literature data and our experiences, do not affect the removal of the particle debris to be studied [19–29].

The chemical compositions of the new and retrieved implants were determined by X-ray fluorescence (XRF, Niton XL3t GOLDD+, Thermo Scientific, Waltham, MA, USA) and inductively coupled plasma atomic emission spectroscopy (ICP-AES, Agilent 720, Agilent Technologies, Santa Clara, CA, USA) chemical analyses. The results of the measurements (Ti 87.8, Al 5.6, Nb 6.4, Fe 0.21, Cr 0.007, Ta 0.02 in mass %) were in good agreement with the requirements of ASTM F1295-05 Standard Specification for Wrought Titanium-6 Aluminum-7 Niobium Alloy for Surgical Implant Applications (UNS R56700) [30].

From the new and retrieved Ti6Al7Nb stems we cut the samples for further investigation. For the microstructure analyses and surface chemistry measurements the samples were ground with SiC 220 grinding paper (1 min), polished with MD Largo 9 μm blue lubricant (5 min), and oxide polished with MDCHEM OP-S (STRUERS GmbH, Zweigniederlassung Austria) and $H_2O_2$ (10 min). The samples for the bulk microstructure analyses were additionally etched with Kroll's reagent (www.astm.org/Standards/E407.htm).

Samples for the investigation of the surface properties (wettability, roughness) and morphology using scanning electron microscopy (SEM) and energy-dispersive X-ray spectroscopy (EDS) were prepared from the new and retrieved stems, cleaned in an ultra-sonic bath of isopropyl alcohol and dried in dry nitrogen gas. The samples for the surface-chemistry analyses of the passive film had polished surfaces.

## 2.2. Wettability

The static-water contact-angle measurements on the as-received and sterilized stems were performed at three characteristic sites (at the root, in the middle and at the neck) using a surface-energy evaluation system (Advex Instruments s.r.o.). Water droplets of 5 μL were deposited on the surface and the average contact angle was determined using Young–Laplace fitting from at least five measurements on each characteristic site. The measurements were carried out at 20 °C and an ambient humidity of 55%.

An optical 3D metrology system, Alicona Infinite Focus (Alicona Imaging GmbH, Raaba, Austria), was employed for the surface-roughness measurements on the as-received and sterilized stems at three characteristic sites (at the root, in the middle and at the neck). Three measurements were performed at each site using a magnification of 20× with a lateral resolution of 0.9 μm and a vertical resolution of about 50 nm. Subsequently, the IF-MeasureSuite (Version 5.1) software was used for an evaluation of the average surface roughness, $S_a$:

$$S_a = \frac{1}{L_x}\frac{1}{L_y}\int_0^{L_x}\int_0^{L_y}|z(x,y)|dxdy \tag{1}$$

where $L_x$ and $L_y$ are the $x$ and $y$ acquisition lengths of the surface and $z(x, y)$ is the height. The size of the analysed area was 714 μm × 542 μm.

An oxygen gaseous plasma treatment was performed on the most hydrophobic hip A-S-16; it was treated in a large RF plasma reactor composed of a glass cylinder with a diameter of ~30 cm (Figure 1a). The hip was treated in an oxygen plasma at a pressure of 50 Pa. The current through the coil was set to 0.3 A. The hip was treated in the plasma for 10 min. After the plasma treatment, the hip was left in the plasma reactor for another 10 min to cool down the surface.

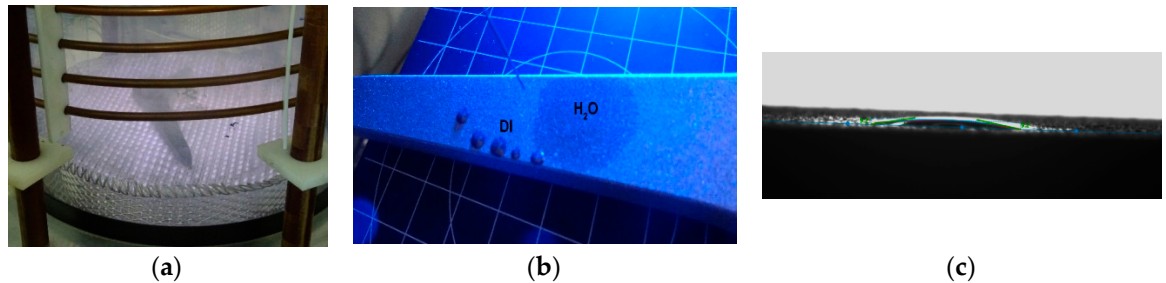

| (**a**) | (**b**) | (**c**) |

**Figure 1.** (**a**) Hip A-S-16 during plasma treatment in a plasma reactor, (**b**) drops of diiodomethane (DI) on the surface of the plasma-treated hip A-S-16. Water completely wetted the surface and did not form a drop (a large dark spot on the surface, (**c**) a water drop on the surface of the hip A-S-16 one hour after plasma treatment.

## 2.3. Scanning Electron Microscopuy (SEM) Analysis

For the morphology, microstructure and chemistry the samples were analysed using a field-emission scanning electron microscope (ZEISS crossbeam 550 FIB-SEM, Carl Zeiss AG, Oberkochen, Germany). The instrument is equipped with secondary-electron (SE) and backscattered-electron (BE) imaging modes for analyses of the morphology of the samples and EDS (EDAX, Octane Elite, Draper, Cambridge, MA, USA) for the surface chemistry. For the SE and BE imaging an acceleration of 15 kV at a current of approximately 0.5 nA was used at a vacuum below $10^{-6}$ mbar.

### 2.4. Auger Electron Spectroscopy (AES) and X-ray Photoelectron Spectroscopy (XPS) Surface Analysis

2.4.1. AES Analysis

AES was used to determine the composition and the thickness of the native thin surface-oxide films on the polished implant surface. A Microlab 310-F spectrometer (Thermo Scientific, Waltham, MA, USA), described in our previous studies [5] was used. The parameters of the AES analyses included a 10 keV primary electron beam at a current of 1 nA an angle of incidence of 0°, and an Auger emission angle of 30°. AES depth profiling was performed by argon-ion sputtering at 3 keV and scanning the ion beam over a 2 mm × 2 mm area. The sputtering rate was about 0.7 nm/min. The AES spectra were acquired using Avantage 3.41 v data-acquisition and data-processing software supplied by Thermo Scientific, [31–33]. In addition, Casa X-ray photoelectron spectroscopy (XPS) software (http://www.casaxps.com/) was employed.

2.4.2. XPS Analysis

The XPS analyses were carried out on the PHI-TFA XPS spectrometer of Physical Electronics Inc. (Chanhassen, MN, USA). Samples were polished before the analyses and exposed to air for one day. The native oxide layer was analysed for thickness and composition on two TiAlNb samples. The analysed area was 0.4 mm in diameter and the analysed depth was about 3–5 nm. This high surface sensitivity is a general characteristic of the XPS method. Sample surfaces were excited by X-ray radiation from a monochromatic Al source at a photon energy of 1486.6 eV. Quantification of the surface composition was performed from the XPS peak intensities, taking into account the relative sensitivity factors provided by the instrument manufacturer [33]. We estimate that the relative error of the calculated concentrations is about 20% of the reported values. In order to analyse the in-depth distribution of elements in the sub-surface region up to 25 nm, XPS depth profiling was performed in combination with ion sputtering. Ar ions of energy 3 keV were used. The velocity of the ion sputtering was estimated to be 1.0 nm/min, calibrated on a Ni/Cr multilayer structure of known thickness.

### 2.5. Corrosion-Resistance Measurements

An electrochemical study was carried out in a simulated physiological Hank's solution at 37 °C and pH = 7.8 using a BioLogic Modular Research Grade Potentiostat/Galvanostat/FRA Model SP-300 (BioLogic Science Instruments, Seyssinet-Pariset, France) with EC-Lab Software (EC-Lab(r) V11.10). The experiments were performed in a three-electrode flat corrosion cell where the working electrode (WE) was the tested specimen, the reference electrode (RE) was a saturated calomel electrode (SCE, 0.242 V vs. SHE) and the counter electrode (CE) was a platinum net. The tested specimens were stabilized for 1 h at the open-circuit potential (OCP) before the electrochemical measurements. The potentiodynamic curves were measured with a scan rate of 1 mV·s$^{-1}$ from −250 mV vs. SCE according to the OCP [34]. All the measurements were repeated three times.

## 3. Results and Discussion

### 3.1. Wettability and Roughness

The surface wettability was analysed using five static contact-angle measurements with water at three characteristic spots on each stem: neck, middle and root. The average contact-angle values at each site, with an estimated error in the reading of θ ± 2.0°, were measured. The average surface-roughness parameter was used to evaluate the morphological characteristics of the stems and the three sites of the measurements.

Contact-angle and average-surface-roughness measurements at the three characteristic sites on a new stem before and after the sterilization are shown in Table 1. We can see that the new stem is moderately hydrophobic, and the wettability is practically the same at all three sites. Sterilization, however, turns the surface of the stem hydrophilic. Interestingly, there is a large difference in the

wettability between the three sites: the root being strongly hydrophilic compared to the middle and the neck.

**Table 1.** Static water-contact angles θ and average surface roughness, $S_a$, of a new Ti6Al7Nb stem (Smith and Nephew) N1 before and after sterilization.

| Sample Implant N1 | $S_a$ [μm] | Before Sterilization θ [°] | After Sterilization θ [°] |
|---|---|---|---|
| Site 1 | 5.209 | 116.3 | 66.5 (49–85) |
| Site 2 | 5.884 | 114.8 | 73.7 |
| Site 3 | 5.570 | 115.9 | 36.8 |

The average surface roughness is of the same order for all three sites and is not affected by the sterilization.

We performed additional surface analyses on the neck, middle and root of the used stems. Contact-angle and average-surface-roughness measurements of the prematurely failed hip endoprostheses due to aseptic loosening, infection and latent infection are shown in Table 2. The results show that there is no noticeable difference regarding the cause of the premature failure for these reasons. All the surfaces are poorly hydrophilic close to the hydrophobic regime and there is no significant difference in the average surface roughness at different sites and on different stems.

**Table 2.** Static water-contact angles θ and average surface roughness, $S_a$, of prematurely failed hip endoprostheses due to: (a) aseptic loosening, (b) infection, and (c) latent infection.

| Sample | | $S_a$ [μm] | θ [°] |
|---|---|---|---|
| a | Neck | 4.3 ± 0.1 | 82.5 ± 1.5 |
| | middle | 4.2 ± 0.3 | 80.4 ± 2.1 |
| | Root | 4.3 ± 0.1 | 81.3 ± 1.4 |
| b | Neck | 4.6 ± 0.2 | 82.1 ± 2.1 |
| | middle | 4.6 ± 0.2 | 86.9 ± 1.3 |
| | Root | 4.7 ± 0.1 | 91.6 ± 1.9 |
| c | Neck | 4.5 ± 0.1 | 80.4 ± 1.9 |
| | middle | 4.5± 0.2 | 70.6 ± 1.8 |
| | Root | 4.4 ± 0.1 | 75.7 ± 1.4 |

After the plasma treatment, the surface of the hip changed from hydrophobic to super-hydrophilic. When applying a water drop to the surface, it completely wetted the surface—no drop was formed on the surface; therefore, it was not possible to measure the contact angle for the water (dark spot in Figure 1). We could only measure the contact angles for diiodomethane, as shown in Figure 1 and Table 3. When calculating the surface energy shown in Table 3, we proposed a contact angle of 1° for the water. It was found that the surface energy of the plasma-treated hip significantly increased.

**Table 3.** Surface wettability of the selected oxygen plasma-treated hip.

| Sample | Contact Angle of Water (°) | Contact Angle of DI (°) | Surface Energy (mN/m) | Dispersive Component (mN/m) | Polar Component (mN/m) |
|---|---|---|---|---|---|
| Used hip A-S 16 | <1° | 29.5 ± 2.9 | 78.5 ± 1.8 | 44.4 ± 1.2 | 34.1 ± 0.7 |

Because the plasma-treated samples are subjected to surface-ageing effects, we also measured changes in the wettability after the plasma treatment. We found that the hip surface remained super-hydrophilic for at least one hour, because the first meaningful contact angle for the water of ~9°

could be measured more than one hour after the plasma treatment. The variation of the water contact angle with ageing time is shown in Table 4 and Figure 2. After about 2 h of ageing, the water contact angle was stabilized at approximately 20°.

**Table 4.** Variation of the water contact angle with ageing time after plasma treatment.

| Ageing Time (min) | Water Contact Angle (°) |
|---|---|
| 60 | 9.03 ± 0.5 |
| 90 | 16.6 ± 2.0 |
| 150 | 19.1 ± 1.4 |
| 180 | 20.0 ± 1.2 |
| 210 | 19.8 ± 0.8 |

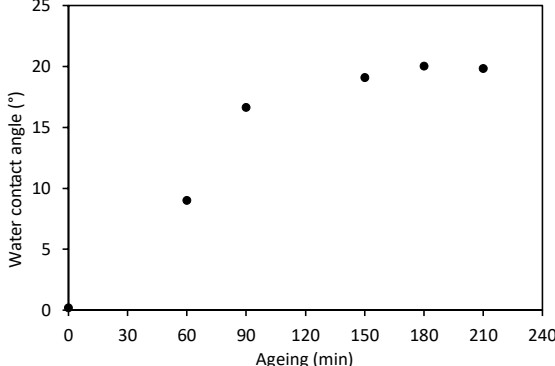

**Figure 2.** Ageing of the Ti6Al7Nb alloy surface after plasma treatment.

We studied two methods of sterilization autoclaving and oxygen plasma sterilization to compare both methods and determined which one is better for sterilization before the application of the primary and tertiary bone cells for the simulation of the osteointegration and for the cell response.

*3.2. Scanning Electron Microscopy (SEM)*

The Ti6Al7Nb microstructure is shown in Figure 3, analysed by SEM/EDS, with the SEI of a two-phase dendrite structure in a matrix of Ti6Al7Nb (Figure 3a) and phase analysis (Figure 3b).

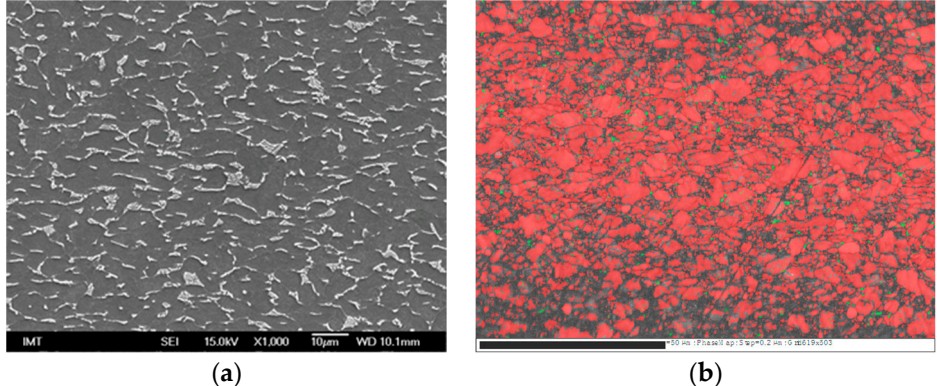

(**a**)            (**b**)

**Figure 3.** (**a**) Secondary-electron image (SEI) and (**b**) electron-backscattered (EBSD) phase image: red Ti close-packed hexagonal crystal structure (hcp), green Ti cubic (bcc) of retrieved Ti6Al7Nb alloy microstructure premature failed due to aseptic loosening.

The microstructures of the new and retrieved stems (premature failed due to aseptic loosening, infection and latent infection) differ slightly. The representative image of the retrieved stem from the Ti6Al7Nb microstructure is shown.

Figure 4a shows the corundum grit-blasted rough surface of a new Ti6Al7Nb alloy stem of an implant and Figure 4b shows EDS mapping of the element distribution of the retained corundum on the rough surface.

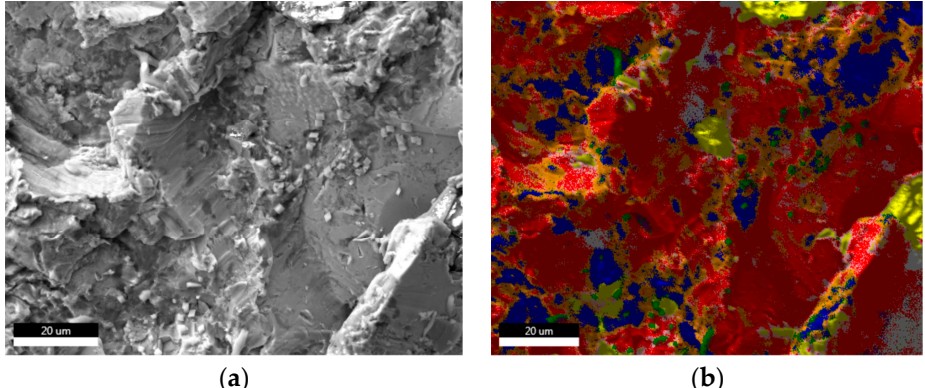

(**a**)                                                                                    (**b**)

**Figure 4.** (**a**) SEI of the corundum grit-blasted surface and (**b**) energy-dispersive X-ray spectroscopy (EDS) mapping showing the retained corundum ($Al_2O_3$) contamination (blue) on the rough surface Ti6Al7Nb (red).

Figure 5a shows a SE image of the cross-section of a new sample, (Figure 5b) EDS mapping of the Ti6Al7Nb matrix (green) and corundum contamination (yellow green). We found that not only is the surface contaminated with retained corundum, but also in the sub-surface there is corundum contamination from the grit-blasted-surface procedure.

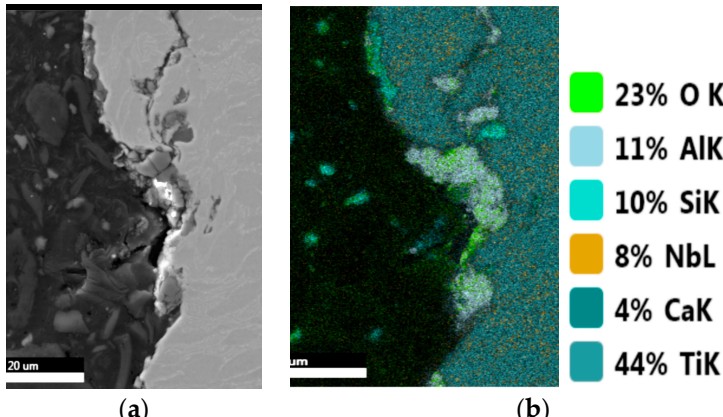

| | |
|---|---|
| 23% O K | |
| 11% AlK | |
| 10% SiK | |
| 8% NbL | |
| 4% CaK | |
| 44% TiK | |

(**a**)                                                                                    (**b**)

**Figure 5.** (**a**) SEI of cross-section sample of new stem, near the surface is corundum contamination (dark grey) from a corundum grit-blasted surface procedure (**b**) EDS mapping showed the Ti6Al7Nb matrix (Ti green) and corundum contamination (Al grey, O yellow-green).

Figure 6a shows a SE image of the cross-section of the sample (I 29) prematurely failed due to infection and (Figure 6b) EDS mapping of the Ti6Al7Nb matrix (yellow) and corundum contamination (black). We found that the surfaces of the retrieved and new stems of Ti6Al7Nb alloy are contaminated with retained corundum. We discovered for the first time the sub-surface corundum contamination from 5 down to 20 μm in depth (Figures 5 and 6).

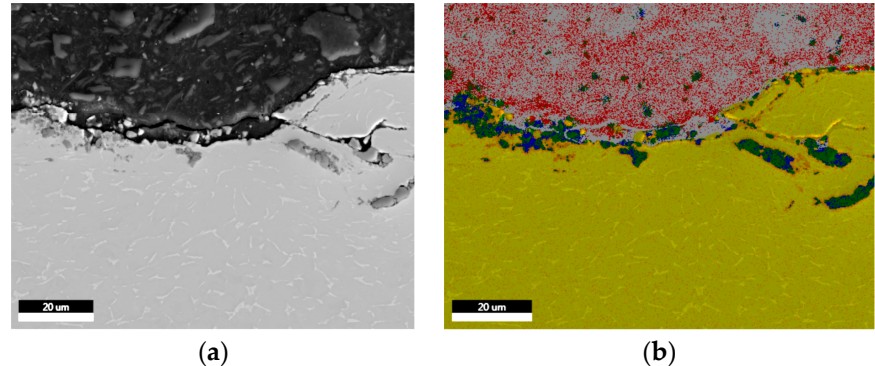

**Figure 6.** (**a**) SEI, cross-section of the sample (I 29) prematurely failed due to infection and (**b**) EDS mapping of the Ti6Al7Nb matrix (yellow) and corundum contamination (black).

The longevity of a total hip prosthesis is limited by the generation of wear debris and its subsequent biological response in periprosthetic tissue [33,35–37]. We found that the surface of new and retrieved Ti6Al7Nb alloy stems are contaminated with retained corundum from the grit-blasted-surface procedure, in good agreement with Rüger et al., up to 20% [36].

Surface corundum contamination is a possible source for aseptic loosening and consequently premature failure.

The main result of our investigations is that we first discovered the sub-surface corundum contamination of new and retrieved Ti6Al7Nb alloy stems, from 5 down to a depth of 20 μm. Corundum wear particles with a hardness of 2000 HV are embedded in a much softer matrix of α + β Ti forged alloy affecting the strengths and the cracks near the corundum particles. These cracks could be adhered by *staphylococcus aureos*, the most prevalent bacteria, which causes implant infection [25–29]. We can conclude that corundum particles could also cause implant infection in addition to aseptic loosening.

The microstructure of the new and retrieved implants is slightly different, but still within the limits of the ASTM F-1295 Ti6Al7Nb alloy standard, which we explain with different charges, different producers and different dates of production [30].

The longevity of total hip prosthesis is limited by the generation of wear debris and its subsequent biological response in periprosthetic tissue. Wear particles induce an adaptive immune response, histologically demonstrated by the presence of lymphocytes within the periprosthetic tissue. This causes the mechanical instability of the joint, then the pain increases due to detrimental biologic responses, osteolysis, component loosening and premature implant failure.

*3.3. Surface Analyses by AES and XPS Measurements*

3.3.1. AES Analysis

The thin native oxide films on the surfaces of the Ti6Al7Nb alloy were analysed by AES. The thicknesses of the thin oxide films on the Ti6Al7Nb (primarily of $TiO_2$) were estimated using AES depth profiling. Ti, O and C Auger peaks were detected in the AES analysis. The estimated oxide thickness was about 5 nm, consisting primarily of $TiO_2$. The amount of $Al_2O_3$ is below the detection limit (<0.1 at.%), with small amounts of Nb oxides also being below the detection limit [31–33].

Figure 7 shows an AES depth profile of the thin oxide films on the surfaces of the Ti6Al7Nb alloy. The $TiO_2$ and traces of $Al_2O_3$ and $Nb_2O_3$ are below the detection limit; the oxide thin film's estimated thickness is 5 nm.

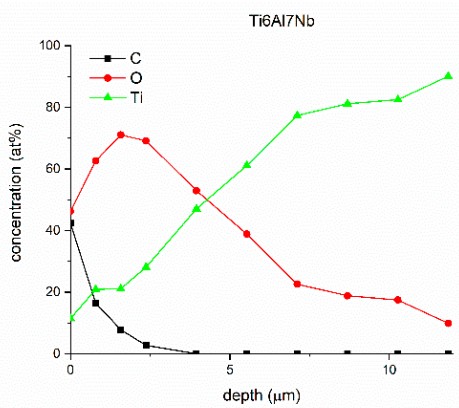

**Figure 7.** Auger electron spectroscopy (AES) depth profile of thin oxide film (passive film) on the surface of the Ti6Al7Nb alloy.

### 3.3.2. XPS Analysis

XPS survey spectra are shown in Figure 8a,c. Traces of Ca were detected on a new sample and traces of Ca and Si were detected on a retrieved sample. The XPS depth profiles are shown in Figure 8b,d. The thickness of the oxide layer on the new alloy was estimated to be 6 ± 2 nm and 5 ± 2 nm on the TiAlNb alloy after application. The oxide layer on both samples consisted of Ti-oxide, Al-oxide and Nb-oxide. These were evidenced by the Ti $2p_{3/2}$ peak at 458.6 eV, characteristic for Ti(4+) in a $TiO_2$-like environment, the Nb $3d_{5/2}$ peak at 207.0 eV, characteristic for the Ni(5+) oxidation state, and the Al $2p$ peak at 74.0 eV, characteristic for the Al(3+) oxidation state.

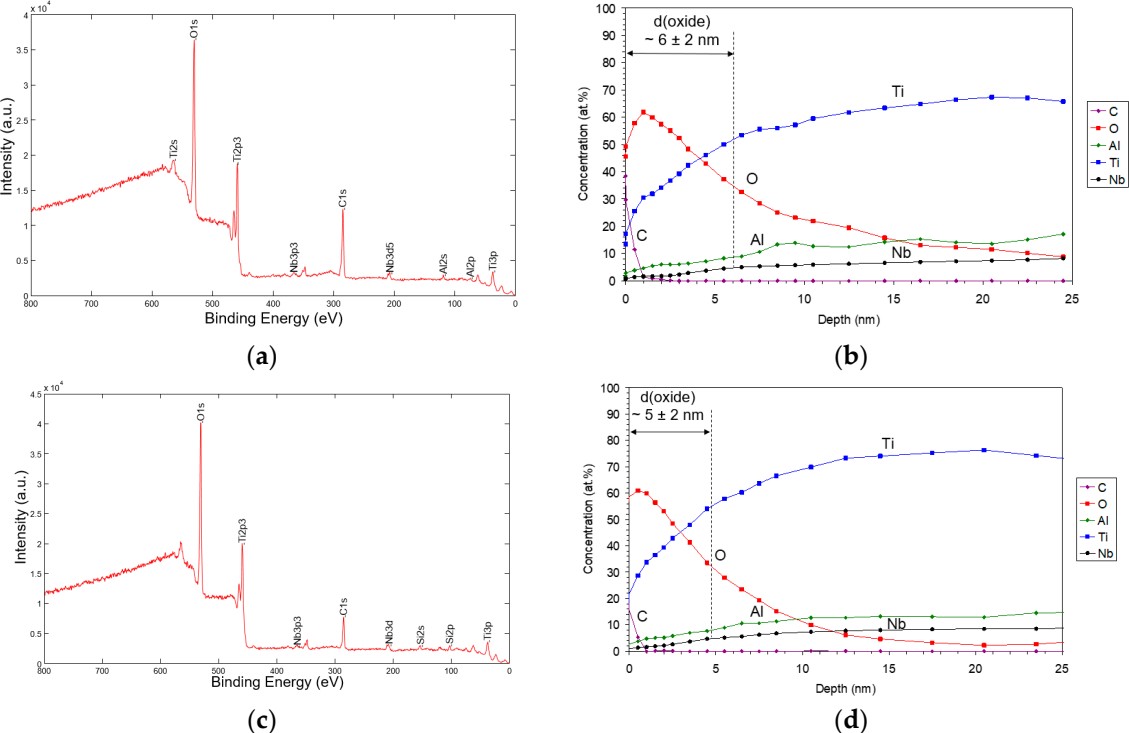

**Figure 8.** (**a**) X-ray photoelectron spectroscopy (XPS) survey spectrum from a new Ti6Al7Nb alloy covered with native oxide after polishing. Traces of Ca were also detected, (**b**) XPS depth profile of a new Ti6Al7Nb alloy covered with native oxide after polishing. Thickness of oxide was estimated to be 6 ± 2 nm, (**c**) XPS survey spectrum of a retrieved Ti6Al7Nb sample covered with native oxide after polishing, traces of Si and Ca were also detected, (**d**) XPS depth profile of a retrieved Ti6Al7Nb alloy covered with a native oxide after polishing. The thickness of the oxide was estimated to be 5 ± 2 nm.

The oxide layer on both samples, new and retrieved, consisted of Ti-oxide, Al-oxide and Nb-oxide. This was evidenced by the Ti $2p_{3/2}$ peak at 458.6 eV, characteristic for Ti(4+) in a $TiO_2$-like environment, the Nb $3d_{5/2}$ peak at 207.0 eV, characteristic for the Ni(5+) oxidation state, and the Al $2p$ peak at 74.0 eV, characteristic for the Al(3+) oxidation state.

In our recent preliminary investigation of retrieved implants that prematurely failed for different reasons (aseptic loosening, infection, latent infection), using the surface-analysis XPS method, we noticed minor differences and proceeded with a detailed investigation of the implant surfaces prematurely failed due to aseptic loosening and latent infection, with the aim of finding possible different mechanisms of premature failure for hip endoprostheses.

### 3.4. Corrosion Resistance

The potentiodynamic behaviour of the investigated materials in a Hank's solution is shown in Figure 9. The quantitative results calculated from the potentiodynamic measurements are presented in Table 5. The calculations of the corrosion rate ($v_{corr}$) and the corrosion-current density ($i_{corr}$) were made according to ASTM G102 – 89 (2015) [34]. The polarization resistance ($R_p$) corresponds to corrosion resistance, which was calculated from the Stern–Geary equation. The corrosion potential ($E_{corr}$) for the new Ti–Al–Nb alloy was approximately −540 mV vs. SCE. A broad range of passivation was observed after the Tafel region. The corrosion rate was calculated at 8.6 μm/year. After different times of exposure to real-body conditions, the samples exhibited different corrosion characteristics depending on the surface condition. Samples I 29, AS 9 and AS_11 exhibited considerably improved corrosion properties with a broad passive region, which were moved to lower corrosion-current densities. The results of this research confirmed that the main role in the corrosion performance of the investigated materials is a compact outer passive film, which inhibits the diffusion of aggressive species and consequently improves the corrosion properties of the material. The observed contamination with $Al_2O_3$ corundum particles on the grit-blasted Ti6Al7Nb surfaces of the new and retrieved implants could lead to increased corrosion due to wear and damage of the protective oxide layer.

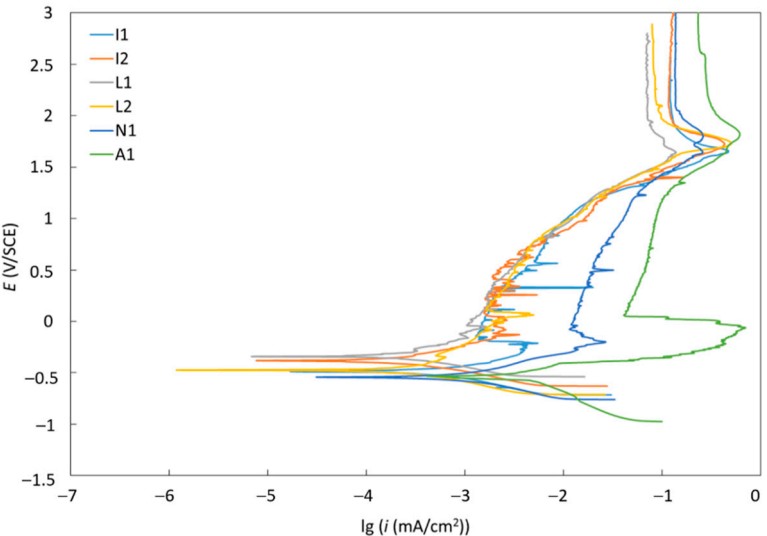

**Figure 9.** Potentiodynamic curves for samples in simulated physiological Hank's solution, pH = 7.8 at 37 °C.

**Table 5.** Electrochemical parameters determined from the potentiodynamic curves of the prematurely failed samples due to: A—aseptic loosening, I—infection, L—latent infection and N—new.

| Material | $E_{corr}$ (mV) | $i_{corr}$ (μm/cm$^2$) | $v_{corr}$ (μm/year) | $R_p$ (kΩ) |
|---|---|---|---|---|
| A1 | −527.41 | 1.18 | 10.26 | 10.5 |
| I1 | −487.62 | 0.87 | 7.56 | 68.7 |
| I2 | −368.87 | 0.24 | 2.10 | 142.7 |
| L1 | −324.74 | 0.15 | 1.32 | 148.2 |
| L2 | −473.57 | 0.32 | 2.80 | 135.8 |
| N1 | −539.60 | 0.98 | 8.52 | 39.2 |

We studied the surface properties, surface chemistry microstructure and corrosion resistance of the new and retrieved hip components that prematurely failed due to aseptic loosening, infection and latent infection. The results showed that there are small differences regarding the microstructure and the surface chemistry between the new and retrieved samples. The roughness is slightly different, while the microstructure, corrosion behaviour and surface chemistry are similar, which is not a surprising result. Only the wettability changes, but this is logical since the surface has been exposed to human fluids, bone cells and bacteria in the human body in the femur canal, a sort of corrosion cell when implanted. It is expected that all the surfaces change from hydrophilic to hydrophobic after some time. What is really interesting is to look for reasons and evidence of infection due to, perhaps, the retained corundum on the sample surfaces and the cracks that occur in the subsurface between the corundum particles and the matrix Ti6Al7Nb alloy from 5 down to a depth of 20 μm. Different bacteria could adhere to these cracks and even form a biofilm [24–28] and could be the source of infection, which will be studied in the near future.

## 4. Conclusions

The longevity of total hip prosthesis is limited by the generation of wear debris and its subsequent biological response in periprosthetic tissue. We found that the surfaces of all the investigated new and retrieved Ti6Al7Nb alloy stems are contaminated with retained corundum from the grit-blasted-surface procedure. In these cases, we confirmed the decreased corrosion performance of the investigated materials due to the observed contamination, which enhances the wear and damage of the protective native oxide layer and, therefore, enables the penetration of aggressive ions and increases the corrosion.

The main result of our investigations is that we first discovered the sub-surface corundum contamination of all the investigated new and retrieved Ti6Al7Nb alloy stems, from 5 down to a depth of 20 μm.

We found cracks near the corundum particles in the sub-surface. Bacteria could adhere to these cracks, causing implant infection [35–38].

We can conclude that the corundum particles possibly induce implant infection, besides aseptic loosening and intergranular or crevice corrosion. The microstructure of the new and retrieved implants is slightly different but still within the limits of the ASTM F-1295 Ti6Al7Nb alloy standard, which we explain with different charges, different producers and different dates of production. In the near future we will investigate these small microstructural differences.

Wear particles cause an adaptive immune response, which is histologically demonstrated by the presence of lymphocytes in the periprosthetic tissue. These cause the mechanical instability of the joint, after which the pain increases with detrimental biological responses, osteolysis, component loosening and premature implant failure.

Our conclusion is that the surface and sub-surface should be free of local corundum contamination. Therefore, the optimization of the corundum grit-blasting procedure of titanium alloys is necessary to minimize the number of residual particles and to develop a new kind of modified nanostructured surface for the implants to achieve better longevity and reduced health costs.

**Author Contributions:** Conceptualization, K.A., D.D., B.K., I.B., and M.J.; Methodology, K.A., A.K., M.C., A.V., I.B., Č.D., J.K., M.G. (Matjaž Godec) and M.G. (Matevž Gorenšek); Formal Analysis, M.J., M.G. (Matjaž Godec), M.G. (Matevž Gorenšek), A.K., J.K., A.V., J.K., K.A., Č.D., B.Š.B. and D.D.; Investigation, K.A., J.K., M.J., M.G. (Matjaž Godec), A.K., M.C. and M.G. (Matevž Gorenšek); Writing—Original Draft Preparation, K.A., D.D. and M.J.; Writing—Review and Editing, M.J., K.A. and M.C.

**Funding:** This research was funded by the Slovenian Research Agency ARRS (P2-0132–Research Program Institute of Metals and Technology) and Tertiary projects of the Department of Orthopedic Surgery of University Medical Center Ljubljana UKCLJ20180128 and UKC20190145.

**Acknowledgments:** The responsible proof reader for the English language is Paul McGuiness, Institute of Metals and Technology, Ljubljana Slovenia.

**Conflicts of Interest:** The authors declare no conflict of interest.

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
