# Peer review of "Surface Properties of Retrieved Cementless Femoral Hip Endoprostheses Produced from a Ti6Al7Nb Alloy"

_coatings, doi:10.3390/coatings9120868_

Round 1
Reviewer 1 Report
1. The manuscript entitled Surface Properties of retrieved cement less femoral hip endoprosteses produced from Ti6A17Nb Alloy is a very interesting study with shows potential causes of premature hip replacement failure.

Author Response
Please, see the attachement

Reviewer 2 Report
1. Excellent manuscript with interesting data and helps with literature and added to the value....
Author Response
Thank you for your valuable opinion
Reviewer 3 Report
The work by Klemen Avsec et al. presents an experimental study of the surface of Ti6Al7Nb alloy femoral stems after retrieval due to failure, and compare these surfaces with new non-implanted ones. The subject of research is very interesting and attractive, worth to be studied. Nevertheless, there are some aspects that need improvement prior to publication.
1. As stated in page 2, the objectives of the work are as follows: “In the present study the surface properties, in terms of wettability, roughness, morphology, microstructure, corrosion and the surface chemistry of the Ti6Al7Nb alloy of retrieved and new femoral components, i.e., the stems, of cementless hip endoprostheses, were studied in detail”. Authors studied the surfaces of retrieved samples and compared them with the surface of new ones. The results show that there are no differences between new and implanted samples. This is the main result of the work. Roughness, morphology, microstructure, corrosion behavior and surface chemistry are the same. This is not a surprising result. Only wettability changes, but this is logical since surface has been “aged” when implanted, and all surfaces change from hydrophilic to hydrophobic after some time due to surface contamination with, for example, C.
2. What is really interesting is to look for reasons and evidences of infection due to, perhaps, the corundum remainings on the samples surface. By this is not carried out in the present work.
3. On the other hand, the paper collects the different surface analyses of the different samples (new and retrieved ones), but there is almost no discussion throughout the manuscript.
4. Why some samples were treated with plasma? What is the purpose of this study?
5. Some conclusions are not based on the results shown in the manuscript. In line 308 it is said: “We found Al2O3 corundum particle contamination on the grit-blasted Ti6Al7Nb surfaces of the new and retrieved implants. In these cases, we confirmed the decreased corrosion performance of the investigated materials due to the observed contamination, which enhances the wear and damage of the protective oxide layer and therefore enables the penetration of aggressive ions and increases corrosion”. Throughout the manuscript there is no evidence of surface wear or damage. Regarding the effect of corundum particles on corrosion, the manuscript shows no evidences of this fact. As shown in Table 5, new and retrieved samples have almost similar corrosion behavior. In order to make this assertion, it would be necessary (and is advisable) to compare non-contaminated by Al2O3 rough Ti6Al7Nb samples, with the samples studied in the present work.
6. Same happens in line 313, it is said: “These residues are a potential problem, i.e., third-body wear particles and probably induction for osteolysis and aseptic loosening. We found that not only is the surface contaminated with retained corundum, but also the subsurface is corundum contaminated from the grit-blasted surface procedure”. This statement may be true, but there is no evidence in the present work of any osteolysis or aseptic loosening due to Al2O3 debris or third-body wear.
7. In line 294 it is said: “We found that sterilization, autoclaving at 121ºC, 1.25 bar, for 20 minutes, affects the wetting properties of the Ti6Al7Nb alloy, turning the initially moderately hydrophobic stem hydrophilic. The analyses of the retrieved stems showed the poorly hydrophilic nature of all the samples, regardless of the failure”. This is something usual in all materials. With time, surface aging tends to transform hydrophilic surfaces into hydrophobic due to surface contamination with C.
Other points to check:
8. In line 40-42 it is said: “The Ti6Al7Nb alloy is the most commonly used implant material for the femoral stem and the acetabular cup of hip endoprostheses, and has been in clinical use since 1986”. This assertion is not completely wright. Ti6Al4V has been the most popular metallic implant materials used in THA. In the last decades Ti6Al7Nb has become a very popular material for its better biocompatibility due to the exchange of V (toxic when released over certain critic levels) by Nb. Please modify this statement. In figure 9, include the information about the axis meaning.
Reviewer 4 Report
This manuscript focus on Surface Properties of Retrieved Cementless Femoral Hip Endoprostheses Produced from Ti6Al7Nb Alloy. There contains some interesting results. The manuscript is clear and logical. However, the manuscript is not is not recommended to publish for the following reasons:
The method used in this manuscript is not innovative enough. In fact, most of the work was carried out in the previous paper. Also the authors need to highlight their innovative contributions. There are many errors in the manuscript, such as, in page 6, line 202, “Table 3 and Figure 1” would be “Table 4 and Figure 3”. The abstract and conclusion are not concise enough. The abstract should highlight the research focus or major findings of the article. In addition, the conclusion section should generally be shortened, not all experimental results should be shown in the conclusion section, and the unimportant results should be omitted. There are too many statistics to explain the meaning of the results. In fact, more emphasis should be placed on the mechanism for discussing these results than on the initial statistical description. In addition, it should be explained why the author did these different experiments. In general, an important article needs to be rewritten, focusing on the important results in the results section, and explaining in the discussion section.
Round 2
Reviewer 3 Report
Authors answered to all queries posed by this reviewer. The manuscript may be accepted for publication.
Reviewer 4 Report
The author gives a reasonable explanation for the questions raised, which can be published at present